# MAD FOR ROBUST REINFORCEMENT LEARNING IN MACHINE TRANSLATION

## ABSTRACT

We introduce a new distributed policy gradient algorithm and show that it outperforms existing reward-aware training procedures such as REINFORCE, minimum risk training (MRT) and proximal policy optimization (PPO) in terms of training stability and generalization performance when optimizing machine translation models. Our algorithm, which we call MAD (on account of using the *mean absolute deviation* in the importance weighting calculation), has distributed data generators sampling multiple candidates per source sentence on worker nodes, while a central learner updates the policy. MAD depends crucially on two variance reduction strategies: (1) a conditional reward normalization method that ensures each source sentence has both positive and negative reward translation examples and (2) a new robust importance weighting scheme that acts as a conditional entropy regularizer. Experiments on a variety of translation tasks show that policies learned using the MAD algorithm perform very well when using both greedy decoding and beam search, and that the learned policies are sensitive to the specific reward used during training.

## 1 INTRODUCTION

There is increasing interest in fine-tuning conditional language models on the basis of feedback from task-specific reward models or similarity functions that compare to human-generated reference outputs rather than relying exclusively on supervised learning (Stiennon et al., 2020; Ziegler et al., 2019; Wu et al., 2018; Paulus et al., 2018; Rennie et al., 2017; Ranzato et al., 2016). Maximising sequence level rewards has several advantages. First, it avoids the apparent conflict between the intuitive importance of "getting the full sequence right" in generation problems and the more conventional token-level cross entropy loss. Second, since a policy trained to maximize rewards is supervised with its own outputs—both good and bad ones—it mitigates issues arising from "exposure bias," in which a learned policy that has been trained only on correct examples has no experience recovering from errors and therefore performs poorly at test time (Ranzato et al., 2016). Third, feedback from (learned) rewards can be a cost-effective strategy for incorporating human preferences about how a system should behave (Stiennon et al., 2020; Christiano et al., 2017).

Unfortunately, fine-tuning policies for generating in complex output spaces, such as language, on the basis of sparse rewards is challenging. Estimating and debugging reliable auxiliary critic/value functions that are needed by many learning algorithms is challenging (Wu et al., 2018; Bahdanau et al., 2017; Nguyen et al., 2017), and commonly used average or batch-level reward baselines (Kreutzer et al., 2017) are poor variance reducers since they are independent of the input, and input difficulty is a strong determinant of reward magnitude.

In this paper, we propose a new distributed policy gradient algorithm (§2) for fine-tuning translation models that addresses these issues. The distributed setup lets us use modest computation to obtain simple and effective empirical reward baselines (Rennie et al., 2017) rather than using inappropriate batch-level statistics or relying on brittle auxiliary value models. Our proposed algorithm has two components designed to make learning from the reward signal more effective: we employ a sampling and reward normalization strategy that encourage batches to contain a mix of both positive and negative rewards for each source sentence and, second, an importance weighting strategy that encourages the algorithm to pay attention to trajectories that are slightly off the current policy. Thus, our algorithm learns from trajectories that are *already* relatively likely under the current policy

(meaning any updates to the policy will be relatively small), while also encouraging continued exploration throughout training by down-weighting trajectories where the model is very confident. This enables the algorithm to make large improvements in reward while taking small, conservative steps to change the behaviour and, also, helps slow down the rate of policy collapse letting the model obtain continued performance improvements over many training steps.

The policies learned using our MAD algorithm produce high-quality translations, even when using greedy decoding. In our main experiments (§3), we use sentence BLEU as the reward and find that the average improvement on held-out test sets over the initial cross entropy trained model is 2.0 BLEU. We also find that the models are less sensitive to the beam search hyperparameters beam size and length normalization. This means we do not need to optimize the length normalization for each dataset and observe almost no performance degradation with larger beam sizes. We confirm that the algorithm learns different policies depending on the reward function used during optimization, with the resulting policies showing reward-specific improvements on held-out data. Finally, we carry out a careful empirical analysis (§4) to better understand the impact and role the various components of the algorithm have on training dynamics and generalization performance.

## 2    ALGORITHM

Our algorithm consists of workers generating training trajectories and rewards, in parallel, from a slightly out-of-date copy of the policy, $\theta_{old}$, and a central learner that updates the current policy, $\theta$. At the beginning of training, both $\theta$ and $\theta_{old}$ are initialized from a pretrained cross entropy model. The data generation algorithm is shown Alg. 1 and the learner in Alg. 2. The learning algorithm has three core components: sampling from a range of temperatures (§2.1), conditional reward normalization on the basis of empirical means and variances (§2.2), and a novel robust importance weighting strategy that focuses learning efforts on samples that are slightly off policy (§2.3). We discuss each of these components in turn.

---

**Algorithm 1** Asynchronous data generator

1: **function** GENERATE($N, \Delta, T_{min}, T_{max}$)
2:     **while** True **do**
3:         $\theta_{old} \leftarrow \theta$   ▷ *Get current global weights*
4:         $(\boldsymbol{x}, \boldsymbol{y}_{ref}) \sim \mathcal{D}_{train}$
5:         **for** $i \in [1, N]$ **do** ▷ *Obtain $\mathcal{Y}_{\boldsymbol{x}}$, $\mathbf{q}$, and $\mathbf{r}$*
6:             $\Delta \leftarrow (T_{max} - T_{min})/(N - 1)$
7:             $T \leftarrow T_{min} + (i - 1) \times \Delta$
8:             $\boldsymbol{y}_i \leftarrow$ SAMPLE($\boldsymbol{x}, \boldsymbol{\theta}_{old}, T$)
9:             $q_i \leftarrow \log p(\boldsymbol{y}_i \mid \boldsymbol{x}_i; \boldsymbol{\theta}_{old})$
10:            $r_i \leftarrow \Delta(\boldsymbol{y}_i, \boldsymbol{y}_{ref})$
11:         **end for**
12:         $\mu_r \leftarrow \frac{1}{|\mathcal{Y}_{\boldsymbol{x}}|} \sum_i r_i$
13:         $\sigma_r \leftarrow \sqrt{\sum_{i=1}^{|\mathcal{Y}_{\boldsymbol{x}}|}(r_i - \mu_r)^2/|\mathcal{Y}_{\boldsymbol{x}}|}$
14:         $\tilde{\mu}_q \leftarrow$ MEDIAN($\mathbf{q}$)
15:         $\tilde{\sigma}_q \leftarrow$ MEDIAN($|\mathbf{q} - \tilde{\mu}_q|$) ▷ *Median dev.*
16:         **for** $i \in [1, |\mathcal{Y}_{\boldsymbol{x}}|]$ **do**
17:             $\bar{r}_i \leftarrow (r_i - \mu_r)/\sigma_r$
18:             $v_i \leftarrow \exp(-|q_i - \tilde{\mu}_q|/\tilde{\sigma}_q)$
19:             ENQUEUE($\boldsymbol{x}, \boldsymbol{y}_i, q_i, \bar{r}_i, v_i$)
20:         **end for**
21:     **end while**
22: **end function**

**Algorithm 2** Learning algorithm

1: **function** LEARN($S, \eta, \boldsymbol{\theta}_{CE}$)
2:     $\boldsymbol{\theta} \leftarrow \boldsymbol{\theta}_{CE}$
3:     **for** $i \in [1, S]$ **do**
4:         $\boldsymbol{x}, \boldsymbol{y}, q, \bar{r}, v \leftarrow$ DEQUEUE()
5:         $p \leftarrow \log p(\boldsymbol{y} \mid \boldsymbol{x}; \boldsymbol{\theta})$
6:         $u \leftarrow \exp(p - q)$
7:         $\alpha \leftarrow \mathbb{SG}(\min\{u \times v, \ 2\})$
8:         $\mathcal{L} \leftarrow \alpha \times \bar{r} \times \log p(\boldsymbol{y} \mid \boldsymbol{x}; \text{dropout}(\boldsymbol{\theta}))$
9:         $\boldsymbol{\theta} \leftarrow \boldsymbol{\theta} + \eta \times \frac{\partial \mathcal{L}}{\partial \boldsymbol{\theta}}$
10:     **end for**
11: **end function**

---

### 2.1    MULTI-TEMPERATURE SAMPLING

To obtain suitably diverse candidates to learn from, it is conventional to add a temperature hyperparameter used to generate samples (Shen et al., 2016; Papini et al., 2020). We identify two problems with this. First, there is the practical matter of needing to select a temperature in order to obtain good performance. Second, it is widely observed that policy gradient algorithms result in increasingly

peaked distributions as training progresses (Kiegeland & Kreutzer, 2021; Choshen et al., 2020; Rennie et al., 2017), meaning that the amount of "exploration" being considered by the algorithm decreases over time. While some RL tasks are interested in maximising total accumulated returns over time (meaning that "exploitation" is important), we rather seek to learn a policy that is capable of behaving as intelligently as possible in states that are not encountered during a training phase, and therefore, we seek an exploration-heavy sampling strategy. To avoid the difficulties with drifting entropy, we use a simple approach of generating populations of samples at several different temperatures.

Concretely our data generation algorithm (lines 5–8 of Alg. 1) begins by sampling $N$ translations for the current policy and source sentence $\boldsymbol{x}$. Each sample is generated using a different temperature that is determined by finding *equally spaced values*[1] between the interval $[T_{min}, T_{max}]$. Duplicate translations are removed. The process we use is:

$$\mathcal{Y}_{\boldsymbol{x}} = \text{UNIQUE}(\{\text{SAMPLE}(\boldsymbol{x}, \boldsymbol{\theta}, t);\ t \in T\})$$
where
$$T = \{T_{min} + \delta_t \cdot (i - 1);\ i \in \{1, \dots, N\}\}$$
$$\delta_t = \frac{T_{max} - T_{min}}{N - 1}.$$

## 2.2 CONDITIONAL REWARD NORMALIZATION

Reward normalization is a well known variance reduction technique for policy gradient methods, with the simplest version being to subtract a constant from the reward (Williams, 1992). Other methods have been proposed such as subtracting a model *baseline reward* (Weaver & Tao, 2001), the running average of the rewards seen during training (Kreutzer et al., 2017), or $z$-scoring the rewards in a training batch (Stiennon et al., 2020). As demonstrated by Kiegeland & Kreutzer (2021), using the running reward average $b$ helps to reduce variance when REINFORCE is applied to translation.

While empirically effective, these baselines explain less reward variation than we might hope. We note that the difficulty of generating a good translation is strongly dependent on the intrinsic difficulty of the source sentence, not merely the current policy (Don-Yehiya et al., 2022). Since the usual reward normalization methods do not take this dependency into account, this results in a bias toward giving difficult sentences negative rewards and easier sentences positive rewards, leading to less stable learning (see §4.2 and Appendix A.3).

We therefore use a standardization method that is conditioned on the source sentence. We take the set of translations for source sentence $\boldsymbol{x}$ and obtain a vector of rewards $r_i = \Delta(\boldsymbol{y}_i, \boldsymbol{y}_{ref}) \ \forall i \in [1, |\mathcal{Y}_{\boldsymbol{x}}|]$ where $\Delta(\boldsymbol{y}, \boldsymbol{y}_{ref})^2$ is a scalar valued reward indicating how well $\boldsymbol{y}$ communicates the contents of the reference translation $\boldsymbol{y}_{ref}$. We then standardize these rewards by removing the mean and dividing by the standard deviation (lines 12–20 of Alg. 1),

$$\overline{r}_i = (r_i - \mu_r)/\sigma_r \ \forall i \in [1, |\mathcal{Y}_{\boldsymbol{x}}|], \text{ where}$$
$$\mu_r = \frac{1}{|\mathcal{Y}_{\boldsymbol{x}}|} \sum r_i, \quad \sigma_r = \sqrt{\frac{1}{|\mathcal{Y}_{\boldsymbol{x}}|} \sum (r_i - \mu_r)^2}.$$

This ensures that every source sentence, irrespective of its translation difficulty, has examples with positive and negative rewards.

In contrast, the standard reward used for the PPO algorithm is the $z$-scored reward of the training batch, $\widetilde{r}_i = (r_i - \widetilde{\mu}_r)/\widetilde{\sigma}_r \ \forall i \in [1, |\mathcal{B}|]$, where $\mathcal{B}$ is the training batch with randomly sampled $(\boldsymbol{x}, \boldsymbol{y})$ examples and $\widetilde{\mu}_r$ and $\widetilde{\sigma}_r$ are respectively the mean and standard deviation of the rewards in $\mathcal{B}$.

## 2.3 MAD IMPORTANCE WEIGHTS

A key feature of our algorithm—and the one that gives it its name—is the use of a new importance weighting scheme for deciding which sampled trajectories are most valuable for updating the policy

---

[1] `https://numpy.org/doc/stable/reference/generated/numpy.linspace.html`

[2] For most of this work $\Delta(\boldsymbol{y}, \boldsymbol{y}_{ref})$ refers to sentence BLEU, see §4.6 for alternatives and analysis.

$$\mathcal{L}_{\text{REINFORCE}} = (r - b) \cdot \log p(\boldsymbol{y} \mid \boldsymbol{x})$$
$$\mathcal{L}_{\text{PPO}} = \min\{u \cdot \widetilde{r}, \ \text{clip}(u, 1 - \epsilon, 1 + \epsilon) \cdot \widetilde{r}\}$$
$$\mathcal{L}_{\text{MRT}} = \sum_{\boldsymbol{y} \in \mathcal{Y}_x} r \cdot \frac{p(\boldsymbol{y} \mid \boldsymbol{x})}{\sum_{\boldsymbol{y}' \in \mathcal{Y}_x} p(\boldsymbol{y}' \mid \boldsymbol{x})}$$
$$\mathcal{L}_{\text{MAD}} = \min\{u \cdot v, \ 2\} \cdot \overline{r} \cdot \log p(\boldsymbol{y} \mid \boldsymbol{x})$$

Figure 1: Sequence level losses used by the algorithms evaluated in this paper. $\boldsymbol{x}$ is the source sentence and $\boldsymbol{y}$ is a sampled translation. Here $r = \Delta(\boldsymbol{y}, \boldsymbol{y}_{ref})$ is the reward, which is sentence BLEU; $\widetilde{r}$ and $\overline{r}$ are different ways of normalizing said reward. The policy sequence probability is $p(\boldsymbol{y} \mid \boldsymbol{x})$ is while $u$ and $v$ are importance weightings that depend on the this value along with the behaviour sequence probability, $q(\boldsymbol{y} \mid \boldsymbol{x})$. See §2 for full explanation.

and which others should be downweighted. For trajectory $i$, our full importance sampling correction is

$$w_i = \min\{u_i \cdot v_i, \ 2\}. \tag{1}$$

We explain $u_i$ and $v_i$ below and note that truncating importance weights is a standard variance reduction strategy (Cortes et al., 2010; Schulman et al., 2017).

The first term in Eq. 1, $u_i$, is the standard importance weighting ratio (Precup et al., 2000; Pang & He, 2021) to deal with the fact that in a distributed setup, data is generated from a stale policy:

$$u_i = \exp(p_i - q_i), \ \text{where}$$
$$p_i = \log p(\boldsymbol{y}_i \mid \boldsymbol{x}; \boldsymbol{\theta}), \quad q_i = \log p(\boldsymbol{y}_i \mid \boldsymbol{x}; \boldsymbol{\theta}_{old}).$$

The second term in Eq. 1, $v_i$, encourages continued exploration during training by having the learner pay attention to samples that are "relatively likely" under the current policy (as approximated by $q$, since we want the calculation to be carried out on the worker nodes, and $p$ is only available on the central learner). We operationalize the notion of "relatively likely" as something that is near to the median[3] probability under $q$ of the elements in $\mathcal{Y}_{\boldsymbol{x}}$, using the exponentiated negative *median absolute deviation* (MAD; lines 14–20):

$$v_i = \exp(-|q_i - \tilde{\mu}_q|/\tilde{\sigma}_q)$$
$$\tilde{\mu}_q = \text{MEDIAN}(\mathbf{q})$$
$$\tilde{\sigma}_q = \text{MEDIAN}(|\mathbf{q} - \tilde{\mu}_q|).$$

Why do we want to concentrate learning on these relatively likely trajectories, rather than perhaps other trajectories that have even higher (or lower) rewards? First, down-weighting very unlikely samples is a strategy for reducing the gradient variance, since gradients associated with smaller changes to the policy will require, on average, less significant updates to the network. This is advantageous since smaller gradient variance can lead to better generalization error Kreutzer et al. (2017). Second, Choshen et al. (2020) provide evidence that RL algorithms cause the policy to become more peaked during training which means that sampling diversity decreases and, thus, the amount exploration as training progresses is reduced (§4.3). Since we seek an exploration-heavy RL algorithm, we focus learning effort on instances which are "in reach" of the current policy but not at the current policy's mode.

## 3 EXPERIMENTS

### 3.1 DATASETS

We run our model along with all the baselines on a total of 9 dataset–language direction translation tasks. See Appendix C for the detailed description of dataset, preprocessing, and tokenization.

---

[3]The median was chosen because $\mathbf{q}$ can have a long tail due to degenerate translations.

## 3.2 TRAINING

For each task, we pretrain a sequence-to-sequence transformer model (Vaswani et al., 2017), using a word level cross entropy loss, until convergence. We refer to this model as the Cross Entropy (CE) model. All treatments for a task are initialized using the same CE checkpoint, which is the checkpoint that had the highest development set BLEU. The treatments are trained using the same training/development sets for a total of 200K steps with early stopping if the average development BLEU is not improved for the last 20K steps. In the case of PPO and MAD, every 20 training steps the learning node saves a checkpoint which the workers and evaluators load asynchronously. We use a global batch size of 256 training examples per step. See Appendix B for detailed hyperparameter configurations and Appendix D for implementation details.

## 3.3 COMPARED RL OBJECTIVES

Figure 1 gives the primary RL objectives we compare. These are: the REINFORCE algorithm (Williams, 1992) using a moving average baseline (Weaver & Tao, 2001), a proximal policy optimization algorithm (Schulman et al., 2017, PPO), minimium risk training (Shen et al., 2016, MRT), and our algorithm MAD. Since REINFORCE and MRT are on-policy algorithms, we optimize these objectives on a single worker; the distributed infrastructure is only used for MAD and PPO.

## 3.4 EVALUATION AND HYPERPARAMETERS

We use the *sacreBLEU* script Post (2018)[4] with detokenized model hypothesis and the original references.[5] When running on the test set, we select the checkpoint that had the highest development set BLEU under greedy decoding.

The important hyperparameters of our algorithm are $N$, $T_{min}$, and $T_{max}$. During preliminary experiments, we swept over $N$ and found that anything in the range of $[8, 24]$ provided similar results. We fixed it at $N = 12$ for all the experiments reported in this work. We found that the optimal temperature range was dataset dependent (§4.5) so performed sweeps for all the models in order for fair comparison. For MAD, we swept over 4 interval ranges, $[T_{min}, T_{max}] \in \{[0.2, 0.6], [0.4, 0.8], [0.6, 1.0], [0.8, 1.2]\}$, while for the baselines we found the single best temperature by sweeping over 10 temperatures in the interval $[0.2, 1.2]$ using increments of $0.1$. See Table 3 in Appendix B for selected temperatures.

## 3.5 MAIN RESULTS

Table 1 presents the performance of our algorithm compared to other objectives (§3.3) on 9 machine translation tasks. We compare these models on both greedy decoding and beam search. As can be seen, our reinforcement learning algorithm outperforms REINFORCE, PPO, and MRT by an average of 1.0 BLEU when using greedy decoding. With beam search decoding, MAD outperforms the strong MRT baseline by 0.5 BLEU on average.

We observe that models trained using the MAD algorithm display a smaller gap in performance between greedy decoding and beam search decoding. The explanation for this is related to obtaining more peaked and possibly smoother output distributions (Dabre & Fujita, 2020). The benefit of this is that it reduces the need for per dataset beam search hyperparameter tuning. Specifically, Meister et al. (2020) show that, when using beam search, it is necessary to tune a length normalization term, $\alpha$, for each dataset in order to prevent the model from generating sentences that are too short or too long. Since the MAD algorithm results in models that are less sensitive to $\alpha$ we are able to use a uniform $\alpha$ on all the datasets without sacrificing substantial performance.

We provide some additional remarks along with more test set results using larger beams and other decoding hyperparameters in Appendix A. The test results in Table 8 show that MAD's performance is fairly insensitive to these hyperparameters in marked contrast to CE (Meister et al., 2020).

---

[4]https://github.com/mjpost/sacreBLEU
[5]See Table 7 in Appendix B for settings.

Table 1: Performance of different algorithms on translation. We report sacreBLEU on the held-out test set between the detokenized model hypothesis and original (not tokenized) references. Note that for IWSLT'14 and NIST both the hypotheses and references were lower-cased to keep comparable with prior works. $\mu$ is the average BLEU across all datasets for each method. MAD outperforms other methods by a large margin when greedy decoding is used. The gap shrinks when beam search is used, but MAD still outperforms the next best method, MRT.

Greedy Decoding

| Model | NIST Zh-En | IWSLT'14 De-En | En-De | WMT'14 De-En | En-De | WMT'20 Zh-En | En-Zh | Ps-En | En-Ps | $\mu$ |
|---|---|---|---|---|---|---|---|---|---|---|
| Cross Entropy | 45.5 | 30.1 | 26.7 | 30.5 | 25.7 | 25.0 | 37.3 | 6.6 | 6.0 | 25.9 |
| REINFORCE | 45.5 | 30.2 | 26.7 | 29.3 | 25.7 | 25.0 | 37.1 | 7.2 | 5.9 | 25.8 |
| PPO | 45.9 | 31.0 | 27.7 | 30.8 | 25.6 | 24.8 | 37.3 | 7.3 | 7.5 | 26.4 |
| MRT | 45.2 | 30.8 | 26.7 | 31.3 | 25.8 | 26.4 | 38.8 | 8.4 | 7.5 | 26.8 |
| MAD | **47.4** | **32.1** | **28.0** | **31.8** | **27.1** | **27.5** | **39.5** | **8.8** | **7.9** | **27.8** |

Beam Search | Beams = 5 | Length Normalization ($\alpha$) = 1.0

| Model | NIST Zh-En | IWSLT'14 De-En | En-De | WMT'14 De-En | En-De | WMT'20 Zh-En | En-Zh | Ps-En | En-Ps | $\mu$ |
|---|---|---|---|---|---|---|---|---|---|---|
| Cross Entropy | 47.6 | 31.4 | 27.9 | 31.8 | 26.4 | 26.4 | 37.5 | 7.0 | 6.0 | 26.9 |
| REINFORCE | 47.5 | 31.4 | 28.0 | 30.6 | 26.3 | 26.3 | 37.8 | 8.1 | 6.0 | 26.9 |
| PPO | 47.5 | 31.3 | 28.2 | 31.7 | 26.3 | 26.2 | 37.5 | 7.7 | 7.6 | 27.1 |
| MRT | 47.0 | 31.9 | 27.9 | 32.0 | 26.6 | 27.4 | 39.3 | 9.2 | 7.9 | 27.7 |
| MAD | **48.2** | **32.4** | **28.3** | **32.2** | **27.3** | **28.0** | **39.9** | **9.4** | **8.1** | **28.2** |

## 4 ANALYSIS AND ABLATION EXPERIMENTS

In this section we investigate various aspects of the MAD algorithm. We use a variety of datasets as to not overfit our analysis to the quirks of a specific one. We start off with a full ablation study (§4.1) followed by a closer inspection of the effects of conditional reward normalization (§4.2) and the MAD importance weights (§4.3). After this, we move on to looking at training stability (§4.4) and MAD's temperature range hyperparameter sensitivity (§4.5). Finally, we discuss the impact of optimizing different rewards (§4.6).

### 4.1 FULL ABLATION

In this section we run an ablation study on the two innovations of the MAD algorithm, the conditional normalized rewards (§2.2) and the MAD importance weights (§2.3). For each experiment we use the distributed and sampling components of our algorithm. In experiments where our conditional reward, $\bar{r}_i$, is not included, we use PPO's batch normalized reward, $\widetilde{r}_i$ and in experiments where the MAD weights, $w_i$, are not included, we use the standard importance weights, $\min\{u_i,\ 2\}$. We start with a baseline setup where both the conditional reward normalization and MAD weights are removed; this is the pink line in Figure 5. Next we add back each of these components separately and then, finally, add both components back which results in our MAD algorithm. We can see from this ablation that the primary driver of our results is the conditional reward normalization and that the MAD weights reduce variance enabling the model to continue learning for more training steps.

### 4.2 CONDITIONAL REWARD EFFECT

Our ablation study shows that using conditionally normalized rewards is the most important part of our algorithm. We hypothesized that due to the heterogeneous nature of source sentence translation difficulty that unconditional normalization schemes would systemically assign negative rewards to difficult source sentence translations. To test this hypothesis, we randomly sampled 10 source sentences from the WMT'20 En $\rightarrow$ Zh training dataset and ran our translation sampling algorithm on them. We then compared the assigned rewards for the translations when using batch normalization vs. conditional normalization. Figure 2 shows that when unconditional normalization is used the sign of the reward is strongly determined by the source sentence rather than the relative quality of the translation and that this issue is mitigated by using conditional normalization. We also note that

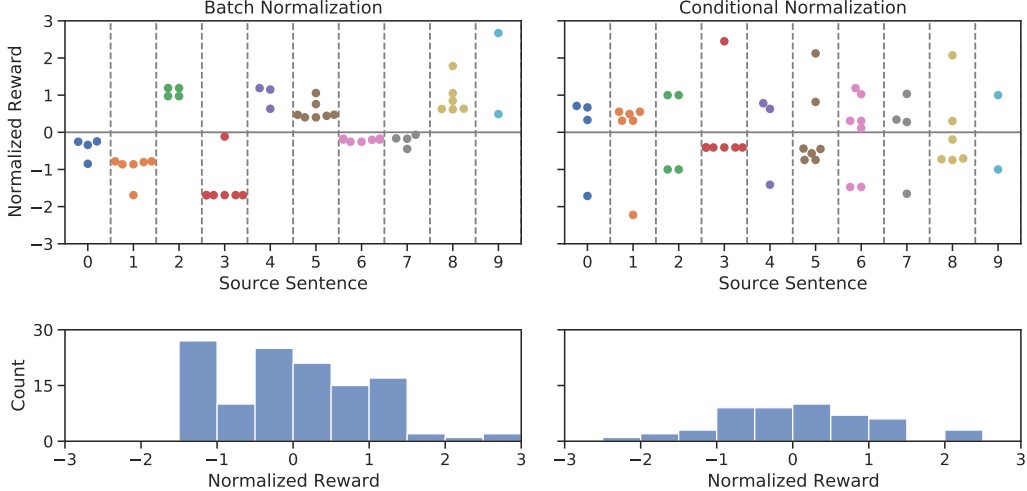

Figure 2: Some source sentences are intrinsically more difficult to translate than others. This results in their translations receiving systematically lower BLEU scores. When unconditional reward normalization such as batch level normalization (*left column*) is applied, the training examples for these difficult source sentences predominately receive negative rewards. To correct this, we introduce conditional reward normalization (§2.2) (*right column*) and see that it produces both positive and negative rewards for every source sentence. We also find that the batch level reward distribution (*bottom row*) is more normal when using conditional normalization.

the distribution of rewards at the batch level is more Gaussian when using conditional normalization. This analysis helps to explain why models trained with an unconditionally normalized reward exhibit a catastrophic drop in performance during training. These models are consistently reducing the entropy of easy source sentences and increasing the entropy of difficult ones which eventually leads to divergence. We explore the concept of source sentence difficulty in the Appendix A.3.

## 4.3 MAD WEIGHTS EFFECT

At first it seems counter-intuitive that we would want to down-weight both high and low probability translation samples. Lowering the weight of very bad translations makes sense because they are off policy and, in many cases, degenerate. However, down-weighting translations that have a very high probability seems counterproductive. Prior work has shown that translation models fine-tuned with policy gradient methods end up having lower entropy predictions (Kiegeland & Kreutzer, 2021; Choshen et al., 2020). We observed that when sampling from a pre-trained cross entropy model, as is done at the very beginning of RL fine-tuning, the highest reward sample is usually the greedily decoded translation which also usually has the lowest entropy since it comes from the mode of the sampling distribution. This results in the policy's mode getting the most positive reinforcement during training and makes the sampling distribution more peaked. We formulated the MAD weights to try and preserve sampling diversity over the course of training and Figure 6 demonstrates that it is effective in this regard. We see MAD weights as a sort of conditional entropy regularizer that focuses learning to occur in the space that is slightly off the current policy. It is conditional in the sense that the level of low entropy penalization (down-weighting) depends on the characteristics of the sampling distribution for the given source sentence and policy state. MAD weights are a step in the right direction but there is more research to be done in this area since it does not prevent policy collapse, just slows down the rate at which it occurs.

## 4.4 TRAINING STABILITY

We look at two aspects of training stability: 1) the amount of variance in generalization performance between random seeds and 2) the rate at which over-fitting occurrs during training. Given this

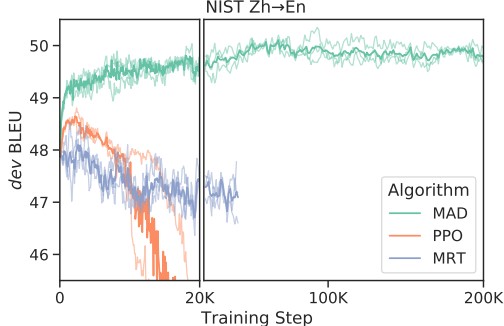
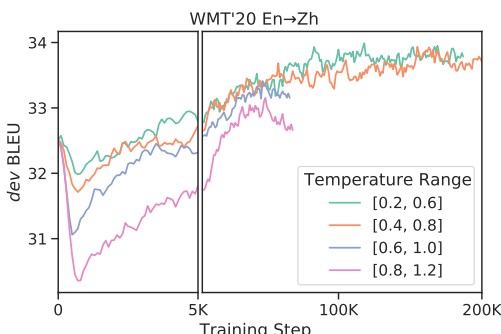

Figure 3: Training curves for 3 random seeds per algorithm. We see that MAD training is stable and has low variance between random seeds. PPO exhibits catastrophic drops in performance while MRT has high random seed variance.

Figure 4: MAD temperature hyperparameter sweep. In line with the other sampling based sequence level optimization baselines we evaluated, MAD is sensitive to the temperature range used to generate samples.

definition of training stability, we can see in Figure 3 that MAD provides stable training for many steps. The difference in generalization performance between the best and worst MAD seed on NIST was 0.15 BLEU. Also, in terms of over-fitting, PPO and REINFORCE consistently produced learning curves similar to the orange lines in Figure 3 across all the datasets we evaluated. We did not observe this catastrophic drop in performance when using MRT or MAD. The primary difference between MRT and MAD in terms of training dynamics is that MAD has lower seed variance and obtains its max development dataset BLEU score later in training.

## 4.5 TEMPERATURE SENSITIVITY

We find that all of the sequence level optimization algorithms evaluated in this work are sensitive to the temperature used during sampling. Figure 4 shows the MAD training curves during a temperature sweep on WMT'20 En $\rightarrow$ Zh. We can see that getting the correct temperature range is important, but that there is some leeway in terms of the optimal range. Table 3 in the Appendix shows the optimal temperatures found for each of the datasets and algorithms. Generally speaking, there is agreement among the various algorithms as to the optimal temperature range for a given dataset. One of the benefits of using a temperature range rather than a single temperature is that a smaller sweep can be performed to find this optimal hyperparameter. We were able to reduce the number of sweep experiments from 10 to 4 (§3.4). The MAD algorithm works just as well when using a single temperature, but we believe the reduction in computational burden during tuning makes using a temperature range worth the effort.

## 4.6 IMPACT OF REWARD TYPE

In the experiments reported so far, we have used sentence BLEU as a reward function. However, Choshen et al. (2020) have shown that uninformative rewards work as well as informative rewards at improving the BLEU score, conjecturing that the reported improvements in these algorithms are due to training artifacts such as reduced entropy of the predictive distribution rather than reward-driven learning. In this section we look at whether training with MAD on different rewards results in a policy that improves those rewards on held-out test sets.[6] We can see in Table 2 that the models are able to generalize on their metric and usually outperform models trained to optimize different metrics. We are able to train a model to optimize multiple metrics and generalize well on all of them.

---

[6]We note that the constant reward used by Choshen et al. (2020) would result in no learning in our algorithm on account of the reward standardisation discussed above in §2.2, so we do not compare to this condition.

Table 2: MAD was used to optimize different rewards on the IWSLT'14 De-En dataset. We report *test* set results for the checkpoint with the max validation performance on the metric being optimized. For TER, a lower score is better, so we optimize $-$TER. The last row, ALL, optimized the equally weighted average of the rewards, $(1/6)(\text{BLEU} + \text{GLEU} + \text{ChrF} + \text{Token F1} - \text{TER} + \text{BLEURT})$.

| Reward Optimized | BLEU | GLEU | ChrF | TER | Token F1 | BLEURT | ALL |
|---|---|---|---|---|---|---|---|
| sBLEU (Papineni et al., 2002) | 32.1 | 30.0 | 55.5 | 50.0 | 55.9 | 58.6 | 30.3 |
| GLEU (Mutton et al., 2007) | **32.2** | **30.5** | 55.7 | 49.1 | 57.0 | 59.5 | 31.0 |
| ChrF (Popović, 2015) | 32.0 | 30.0 | **56.5** | 49.7 | 56.0 | 59.0 | 30.6 |
| $-$TER (Snover et al., 2006) | 32.2 | 30.0 | 55.8 | **48.8** | 56.3 | 59.2 | 30.8 |
| Token F1 | 31.9 | 30.4 | 55.5 | 49.5 | **57.1** | 59.5 | 30.8 |
| BLEURT (Sellam et al., 2020) | 29.5 | 28.4 | 54.9 | 52.0 | 54.8 | **62.3** | 29.6 |
| ALL (*equal weight*) | 32.2 | 30.4 | 56.0 | 49.0 | 56.8 | 59.9 | **31.1** |

# 5 RELATED WORK

During the era of linear translation models, setting parameters to optimize rewards was standard. Although cross entropy is still widely used, Ranzato et al. (2016) inaugurated using RL techniques for neural translation models, and Edunov et al. (2018) provide a thorough survey of the topic and find that RL fine-tuning usually improves the original model. Although it has been shown to optimize a biased approximation to the expected reward objective (Choshen et al., 2020), minimum risk training remains extremely effective (Shen et al., 2016; Kiegeland & Kreutzer, 2021).

More complicated policy gradient methods have been proposed that rely on auxiliary networks. MIXER (Ranzato et al., 2016) predicts a per-token reward baseline while actor-critic methods (Bahdanau et al., 2017) learn a per-timestep Q function concurrently with the translation policy. Although not widely used yet in translation, proximal policy optimization (PPO) (Schulman et al., 2017) has been used to train summarization models from human preferences (Stiennon et al., 2020), and this algorithm provided inspiration for some of the techniques we used here.

Another important source of inspiration in the development of our algorithm was the "hope and fear" online learning algorithm of Chiang (2012), which used a margin-based analysis to argue for a decision rule that selected positive and negative training samples on the basis of a combination of current model score and reward. Additionally, pairwise ranking optimization (Hopkins & May, 2011) reduced the parameter learning problem to classifying pairs of positively and negatively rewarded trajectories, and our objective can be understood as a weighted variant of that objective.

Our work differentiates itself through our conditional reward normalization and approach to entropy regularization through the use of MAD weights.

# 6 CONCLUSION

We introduce a new policy gradient algorithm, MAD, for machine translation and show that it outperforms existing reward-aware training procedures such as REINFORCE, minimum risk training (MRT) and proximal policy optimization (PPO). We test these algorithms on a variety of datasets, language pair directions, and reward functions and demonstrate good test set performance. Our algorithm excels at producing models that generate quality translations, even when greedy decoding is used. Our analysis shows that the conditional reward normalization we introduce is a critical component of the model's success while the MAD weights improve sample diversity and training stability. Taken together, we see an average 2.0 BLEU improvement over the initial cross entropy model and a 1.0 BLEU improvement over the best baseline treatment, MRT. Finally, we offer a method for reducing the number of sweeps needed to find the optimal sampling temperature hyperparameter making it easy to use on new tasks.

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

# A    SUPPLEMENTARY EXPERIMENTS

## A.1    ABLATION STUDY FIGURES

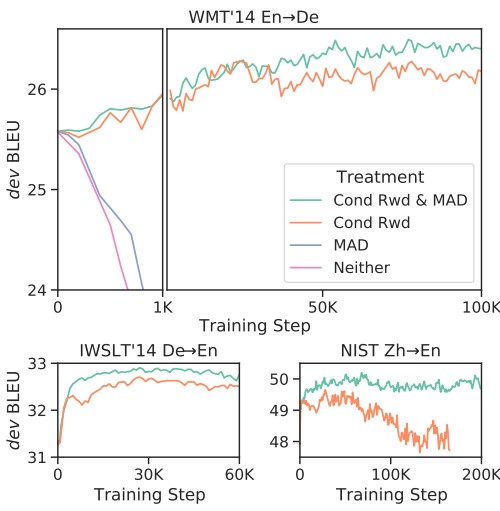

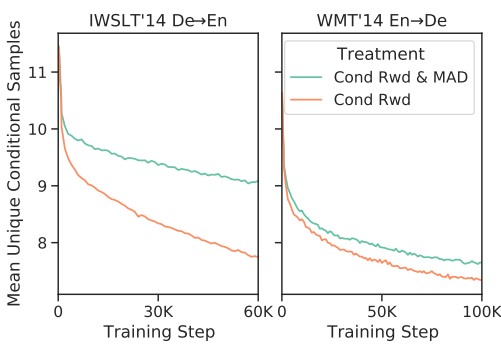

Figure 6: We plot the mean number of unique samples generated by the worker nodes over the course of training for two different datasets. For each source sentence, 12 samples are generated and is thus the upper bound. The green line is the full MAD algorithm while the orange line is an ablation where the MAD weights have been removed. We see that including the MAD weights helps to slow the loss of diversity during training.

Figure 5: Ablation study of the components of the MAD algorithm on multiple datasets, see §4.1 for more detailed description. The most important component of our algorithm is conditional reward normalization, orange lines. Additionally including the MAD weights, green line, improves training stability and offers additional performance improvement.

## A.2    ADDITIONAL BEAM SEARCH RESULTS

Table 8 shows results (on the test set) using larger beams and different decoding hyperparameters. MAD trained with sentenceBLEU continues to outperform other training objectives, and displays the notable behaviour that decoding without length normalization with large beams is effective, whereas performs drops significantly with CE and REINFORCE trained models.

We emphasize that the beam search length normalization term is the same for all models and datasets reported in Table 1. Most prior work in machine translation optimizes $\alpha$ per dataset and only reports this score. We have confirmed that our cross entropy models match prior work when using the tuned $\alpha$. In this sense, we believe the BLEU numbers we report are a lower bound, which might be improved through $\alpha$ tuning.

## A.3    SOURCE SENTENCE DIFFICULTY

It's somewhat difficult to quantify how hard a source sentence will be to translate. However, there is evidence that the uncertainty (as measured by entropy) a powerful Language Model has when modeling the source sentence can be used as a proxy for this (Rae et al., 2022). We see in Figure 7 that there is negative relationship between source sentence entropy and translation performance. This means that unconditional reward normalization methods will incorporate this bias.

## A.4    ADDITIONAL REMARKS

We note that the conditional reward normalization could be applied to both REINFORCE and PPO, however, decided against it to keep these baselines in line with prior work. We also believe our baselines coincide with the form most likely to be used in practice. We did run some preliminary

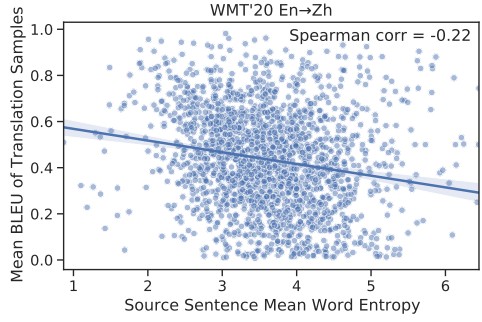

Figure 7: Plot of the relationship between source sentence difficulty and the average score of sampled translations. We use the average word entropy from a strong 280 billion parameter language model (Rae et al., 2022) as a proxy for the uniqueness/difficulty of a given source sentence. We can see that the more uncertainty a language model has about the source sentence, the lower the average BLEU of its sampled translations.

experiments with PPO where we replaced $\widetilde{r}$ with $\overline{r}$, however, the results were mixed. PPO has a few additional hyperparameters that can be tuned so we leave this as future research.

Table 3: Temperatures used for models in main results. REINFORCE used same temperatures as PPO. Only MAD uses a temperature range, other algorithms use as single temperature.

Temperatures

| Dataset | Algo | $T_{min}$ | $T_{max}$ |
|---|---|---|---|
| IWSLT'14 De → En | PPO | 1.0 | |
| | MRT | 0.9 | |
| | MAD | 0.8 | 1.2 |
| IWSLT'14 En → De | PPO | 1.2 | |
| | MRT | 0.6 | |
| | MAD | 0.8 | 1.2 |
| WMT'14 De → En | PPO | 0.2 | |
| | MRT | 0.6 | |
| | MAD | 0.4 | 0.8 |
| WMT'14 En → De | PPO | 1.2 | |
| | MRT | 0.4 | |
| | MAD | 0.4 | 0.8 |
| WMT'20 Ps → En | PPO | 1.2 | |
| | MRT | 1.2 | |
| | MAD | 0.8 | 1.2 |
| WMT'20 En → Ps | PPO | 1.2 | |
| | MRT | 1.0 | |
| | MAD | 0.4 | 0.8 |
| NIST Zh → En | PPO | 1.0 | |
| | MRT | 0.8 | |
| | MAD | 0.6 | 1.0 |
| WMT'20 Zh → En | PPO | 0.8 | |
| | MRT | 0.8 | |
| | MAD | 0.4 | 0.8 |
| WMT'20 En → Zh | PPO | 0.8 | |
| | MRT | 0.6 | |
| | MAD | 0.2 | 0.6 |

Table 4: Base settings for our Transformer models.

Transformer - Base Settings

| Setting | Value |
|---|---|
| embeddings dim | 512 |
| feed forward | 2048 |
| num layers | 6 *enc* + 6 *dec* |
| num heads | 8 |
| dropout | .1 |
| tied embeddings | True |
| tied softmax | True |
| optimizer | Adam |
| learning rate ($\eta$) | 1e-5 |
| $\eta$ warmup steps | 1000 |
| $\eta$ scheduling | Constant |
| gradient clipping | global_norm(1) |
| training steps | 200,000 |
| seq length | 128 |
| global batch size | 256 |

Table 5: Deviations from Transformer base settings.

Transformer - Deltas

| Dataset | Setting | Value |
|---|---|---|
| WMT'20 Ps ↔ En | seq length | 64 |
| | dropout | .3 |
| WMT'14 De ↔ En | seq length | 144 |
| IWSLT'14 De ↔ En | feed forward | 1024 |
| | num heads | 4 |
| | dropout | .3 |

Table 8: Additional test set results when using different Beam Search hyperparameters. Even when large beams are used without length normalization, MAD performs well indicating that it is better calibrated.

Beam Search | Beams = 5 | Length Normalization ($\alpha$) = None

| Model | NIST | IWSLT'14 | | WMT'14 | | WMT'20 | | | | $\mu$ |
|---|---|---|---|---|---|---|---|---|---|---|
| | Zh-En | De-En | En-De | De-En | En-De | Zh-En | En-Zh | Ps-En | En-Ps | |
| Cross Entropy | 47.0 | 31.1 | 27.5 | 31.0 | 26.8 | 25.1 | 34.7 | 7.8 | 5.9 | 26.3 |
| REINFORCE | 47.0 | 31.1 | 27.5 | 29.9 | 26.7 | 25.0 | 34.9 | 7.6 | 5.8 | 26.2 |
| PPO | 47.1 | 31.2 | 28.0 | 31.2 | 26.7 | 25.0 | 34.7 | 8.1 | 7.5 | 26.6 |
| MRT | 47.4 | 31.7 | 27.5 | 31.8 | 27.0 | 27.0 | 38.8 | 9.2 | 7.8 | 27.6 |
| MAD | **48.3** | **32.4** | **28.2** | **32.0** | **27.4** | **27.7** | **39.7** | **9.3** | **8.0** | **28.1** |

Beam Search | Beams = 50 | Length Normalization ($\alpha$) = None

| Model | NIST | IWSLT'14 | | WMT'14 | | WMT'20 | | | | |
|---|---|---|---|---|---|---|---|---|---|---|
| | Zh-En | De-En | En-De | De-En | En-De | Zh-En | En-Zh | Ps-En | En-Ps | |
| Cross Entropy | 42.8 | 28.1 | 26.5 | 28.5 | 24.9 | 22.9 | 32.3 | 4.0 | 1.3 | 23.5 |
| REINFORCE | 43.0 | 28.0 | 26.4 | 28.8 | 25.4 | 22.8 | 32.0 | 2.5 | 1.3 | 23.3 |
| PPO | 45.9 | 31.2 | 28.0 | 29.1 | 24.9 | 23.1 | 32.3 | 6.9 | 6.3 | 25.3 |
| MRT | 47.1 | 30.8 | 26.5 | 31.2 | 27.1 | 26.8 | 38.4 | **9.0** | **7.1** | 27.1 |
| MAD | **48.5** | **32.4** | **28.1** | **31.9** | **27.5** | **27.4** | **39.5** | 7.9 | 6.4 | **27.7** |

Beam Search | Beams = 50 | Length Normalization ($\alpha$) = 1.0

| Model | NIST | IWSLT'14 | | WMT'14 | | WMT'20 | | | | |
|---|---|---|---|---|---|---|---|---|---|---|
| | Zh-En | De-En | En-De | De-En | En-De | Zh-En | En-Zh | Ps-En | En-Ps | |
| Cross Entropy | 48.2 | 31.6 | 27.9 | 31.7 | 25.3 | 25.9 | 36.8 | 5.4 | 2.9 | 26.2 |
| REINFORCE | 48.3 | 31.6 | 28.0 | 30.5 | 26.0 | 25.9 | 36.9 | 6.2 | 2.8 | 26.2 |
| PPO | 47.8 | 31.3 | 28.2 | 31.7 | 25.3 | 25.9 | 36.9 | 7.1 | 6.9 | 26.8 |
| MRT | 47.6 | 32.1 | 27.9 | 31.9 | 26.4 | 27.5 | 39.0 | **9.0** | 7.2 | 27.6 |
| MAD | **48.3** | **32.5** | **28.4** | **32.1** | **27.4** | **28.0** | **39.8** | 8.8 | **7.4** | **28.1** |

# B   HYPERPARAMETERS

We provide the hyperparameters for our Transformer models in Tables 4 and 5. The algorithm specific settings are in Table 6. Finally, the setting used when calculation *sacre*BLEU are found in Table 7.

Table 6: Settings for sequence level algorithms. Note that we did run experiments with $|\mathcal{Y}_x| = 12$ for MRT and found no improvement in development set max BLEU. Given the slow training speed of MRT, we opted to use the convention of 5 samples.

| Algorithm Settings | | |
|---|---|---|
| Algorithm | Setting | Value |
| PPO | $\epsilon$ | 0.2 |
| MRT | $|\mathcal{Y}_x|$ | 5 |
| | max times sampled | 1 |
| MAD | min size to sample | 512 |
| *reverb* | max size | 4096 |
| *settings* | sampler | Uniform |
| | remover | Fifo |

Table 7: *sacre*BLEU settings for each language.

| *sacre*BLEU Settings | | |
|---|---|---|
| Language | Setting | Value |
| De | tokenizer | intl |
| En | tokenizer | 13a |
| Ps | tokenizer | intl |
| Zh | tokenizer | zh |
| All | smooth method | exp |
| | smooth value | 0 |

## C  DATASETS

We evaluate all the models on the following translation tasks: NIST[7] Open MT Chinese → English task, IWSLT'14[8] English ↔ German translation task, WMT'14[9] English ↔ German news translation task, and the WMT'20[10] Chinese ↔ English and Pashto ↔ English news translation tasks. The sizes of each dataset is avaliable in Table 9.

**Preprocessing**  We perform text normalization on the datasets before tokenization.

- All languages - Unicode canonicalization (NKFD from), replacement of common multiple encoding errors present in training data, standardization of quotation marks into "directional" variants.

- English - Replace non-American spelling variants with American spellings using the aspell library.[11] Punctuation was split from English words using a purpose-built library.

- Chinese - Convert any traditional Chinese characters into simplified forms and segment into word-like units using the Jieba segmentation tool.[12]

**Tokenization**  We encode text into sub-word units using the `sentencepiece` tool Kudo & Richardson (2018). When generating our own subword segmentation, we used the algorithm from Kudo (2018) with a minimum character coverage of 0.9995.

Table 9:  Number of training, development, and test examples in each dataset.

| Dataset | Train | Dev | Test |
|---|---|---|---|
| IWSLT'14 De → En | 164K | 7,466 | 6,750 |
| IWSLT'14 En → De | 173K | 1,474 | 6,750 |
| WMT'20 Ps ↔ En | 516K | 5,860 | 2,719 |
| NIST Zh → En | 1.45M | 1,664 | 5,146 |
| WMT'14 De ↔ En | 4.7M | 3,000 | 3,003 |
| WMT'20 Zh → En | 21.8M | 2,000 | 2,000 |
| WMT'20 En → Zh | 21.8M | 1,997 | 1,418 |

Table 10:  We show the training speed as a function of the number of 2x2 TPU workers used. Speed is measured in seconds per 1000 training steps.

| Training Speed | | |
|---|---|---|
| Algorithm | Workers | Sec per 1K |
| REINFORCE | n/a | 2,038 |
| MRT | n/a | 1,728 |
| PPO & MAD | 2 | 1,898 |
| | 4 | 946 |

## D  IMPLEMENTATION DETAILS

**Software**  We use Launchpad (Yang et al., 2021) to create and launch our DAG computing graph. The graph consists of a Reverb table (Cassirer et al., 2021) that holds training examples, multiple worker nodes that generate and publish examples, a learner node that pulls and trains on examples, and an evaluator node that loads fresh checkpoints to calculate dev set sacreBLEU. It is important to not make the Reverb table cache too large to avoid the table filling with stale training examples. This can happen if the training node job is temporarily deallocated while the workers are still running.

**Compute**  All of our program nodes run on a 2x2 TPU configuration. Our hyperparameter sweeps for PPO and MAD used 2 data generating workers running for 50K steps, while final results used 4 workers and ran for 200K steps. Table 10 in the body of the paper provides the wall clock training speed for each algorithm.

---

[7]https://www.nist.gov/itl/iad/mig/open-machine-translation-evaluation
[8]https://sites.google.com/site/iwsltevaluation2014/mt-track
[9]http://statmt.org/wmt14/translation-task.html
[10]http://www.statmt.org/wmt20/translation-task.html
[11]http://wordlist.aspell.net/varcon-readme/
[12]https://github.com/fxsjy/jieba

**Training time** Because of our distributed algorithm, we are able to obtain faster training times by increasing the number of workers that sample from the current policy (this in contrast to purely "on-policy" REINFORCE and MRT algorithms). Table 10 shows that we get near linear scaling as the number of workers is increased.

