# OpenReview forum: "MAD for Robust Reinforcement Learning in Machine Translation"
_ICLR.cc/2023/Conference — Submitted to ICLR 2023_

### Official Review · Reviewer_VmFP · 2022-10-25

**Confidence:** 4
**Clarity, Quality, Novelty And Reproducibility:** See S&W section.
**Correctness:** 2
**Technical Novelty And Significance:** 2
**Empirical Novelty And Significance:** Not applicable
**Recommendation:** 3

**Strength And Weaknesses:**

Strengths:

- Well written in general. Ablations over using conditional reward normalization (fig. 5), MAD IWs (fig. 5,6), and different reward functions (Table 2) are included.

- Performance comparisons with several important baseline methods, including minimum risk training (MRT) (with serious caveats, see limitations below for details).

Limitations:

- Conditional reward normalization (CRN) is identified as the key factor to MAD performance. However, based on the current manuscript (e.g. section 2.2 and discussions elsewhere), it seems that the baseline approaches being compared to such as REINFORCE and MRT, which trivially have conditional variants of their baselines, are NOT conditionally normalized. The importance of CRN is well known, and the comparisons must be "apples to apples".

- Techniques such as MRT and MAD rely on sample population statistics for reward normalization, whereas other techniques estimate a conditional baseline and are much more sample-efficient to train, including cited methods in the paper such as in MIXER (learned baseline), self-critical training (MAP sequence estimate defines baseline), and actor-critic methods for sequence prediction (word slot-dependent rewards). Since conditional reward normalization seems to be the dominating factor in performance, it seems essential to also situate the results relative to one or more of these techniques (the former 2 are very easy to try out).

- Importance weighting based on how far from the median probability each sequence is in probability seems excessively conservative for noisy models such as text sequence generators, and will certainly foster the stated goal of "exploration-heavy" training, but it will also prevent the model from converging. Since this a heurstic, I feel that some other competing heursitics should be compared to and reported on... For example, perhaps the distance from expected probability is better, it is less "exploration heavy", but will allow the policy to converge (for better or for worse).

- Related, it seems that MAD reweighting should be compared with standard regularization baselines, like an entropy term!

- The main components of the approach 1) mean and variance normalization of conditional rewards, and 2) importance weights based on the distance from the median probability can be considered standard and ad hoc, respectively. From this perspective, the paper is lower in novelty for a method-focused ML conference.

**Summary Of The Paper:**

The authors present a simple, distributed approach to robust RL coined MAD. The main components of MAD are:
1) conditional mean and variance normalization of the task reward based on a population of conditional samples, sampled from the current (local) policy over a range of temperatures. This promotes balance in terms of ensuring that there is positive and negative feedback for each set of conditional samples, and regularity in the gradients across sets of conditional samples.
2) An importance weight that depends on the mean absolute deviation (MAD) of the probability of conditional samples from the median probability in a sample population. This is meant to promote more conservative updating of the policy, and mitigate against policy collapse, acting as an entropy regularizer.

Machine translation results on NIST, IWSLT’14, WMT’14, and WMT’20 seem to indicate that the method outperforms several baselines, including MRT (see S&W section for caveats).

**Summary Of The Review:**

Based on the current manuscript, it appears that the proposed method is utilizing conditional baselines, while competing methods which support conditional baselines are not, which is not appropriate, and will necessitate a major revision of the paper before it can be considered for acceptance. In addition, further investigations around their importance weighting heuristic and comparisons with techniques that directly estimate their baselines and so do not need a sample population to train (and so are much more efficient), should also really be included in their experimental investigation.

---

### Official Review · Reviewer_qbvg · 2022-10-26

**Confidence:** 4
**Correctness:** 3
**Technical Novelty And Significance:** 3
**Empirical Novelty And Significance:** 3
**Recommendation:** 5

**Clarity, Quality, Novelty And Reproducibility:**

The paper is clear in describing the techniques, but could be improved in discussion and analysis. Technically, I believe this is a novel interesting contribution.

**Strength And Weaknesses:**

# reasons to accept:
-  This paper is well organized and easy to understand.
- The MAD importance weight is a sophisticated trust region scheme that achieves robust translation against decoder’s beam-size and length normalization. Experiments show that MAD also achieves stable convergence compared to traditional PPO. The scheme also strikes a balance between exploration and optimization, which is worth noting.
- The proposed methods greatly boost stability for RL tuning. The improvement is satisfying compared to the baseline.
- Authors conduct thorough experiments across different languages and different reward types to validate the proposed methods. The results are promising.

# reasons to reject:
- The proposed reward normalization is a standardization, which can hardly be considered as ‘source conditioned’. The authors do not explain how the conditional reward standardization relates to the difficulty of source sentences for its designed purpose.
- According to the ablation study, both MRT and PPO fail to converge on the long run. But SOTA RL tuning involves certain implementation to achieve tuning by MRT and PPO, which should be clarified in baseline settings.
- The paper is somewhat misleading. According to their ablation study, conditional reward normalization is the most important one among the three components. The MAD method, as stated in the title, does not contribute much to the performance gain, and it even fails the training process.
- Reinforcement learning can be tricky and empirical. The tradeoff to implement such methods for tuning rather than data augmentation is worth discussing.

**Summary Of The Paper:**

This paper propose an easy-to-implement fine-tuning for machine translation via reinforcement learning.
The major issue for reinforcement fine-tuning is described as variance reducing, both in rewards and updates.
The main contributions are listed as the following:
- A conditional reward normalization to reduce variance;
- A novel robust importance weighing scheme similar to trust-region update, striking the balance between exploration and exploitation.
The translations are first sampled for tuning rewards via a variable temperature. Then, the rewards are standardized to update the central policy by rescaled weights.
The methods improves the specific translation metric and certain robustness.

**Summary Of The Review:**

The paper presents a interesting framework for fine-tuning MT system. But the analysis and discussion of the method is problematic. It seems to me that the interpretation from the authors are not well-matched with the experiment results.

---

### Official Review · Reviewer_Gvqo · 2022-10-27

**Confidence:** 3
**Correctness:** 3
**Technical Novelty And Significance:** 3
**Empirical Novelty And Significance:** 3
**Recommendation:** 5

**Clarity, Quality, Novelty And Reproducibility:**

This is an original work with good clarity. In my opinion, its quality is around the acceptance threshold of the conference.

**Strength And Weaknesses:**

The strengths of this paper include:
1.	The proposed method shows good performance, especially on greedy decoding.
2.	Very extensive experiments on multiple datasets and multiple settings.
3.	This paper brings new insights for improving RL-Based NMT.

The weaknesses of this paper include:
1.	Figure 5 shows that the major improvements come from conditional reward normalization, which is more of a trick and not novel to me. The improvement of MAD itself seems to be marginal.
2.	Table 2 lacks the results of the cross-entropy baseline. It is important to know whether the proposed method only improves BLEU and lower other metrics.

A few questions:
1.	Will you release the source code?
2.	In REINFORCE and PPO, did you sample multiple translations and calculate their average reward as the reward baseline?


**Summary Of The Paper:**

This paper improves the policy gradient algorithm with three methods: (1)multi-temperature sampling, generate multiple translation samples of the same source sentence using different temperatures; (2)conditional reward normalization, standardize the rewards of the same source sentence by removing the mean and dividing by the standard deviation; (3)MAD importance weights, decide which sampled trajectories are most valuable for updating the policy and which others should be downweighted. Experiments on multiple machine translation datasets show the effectiveness of the proposed method.

**Summary Of The Review:**

This paper improves the policy gradient algorithm with three methods. The performance looks good, but the major improvements come from conditional reward normalization, which is more of a trick and not novel.

---

### Decision · Program_Chairs · 2023-01-20

**Decision:**

Reject

**Justification For Why Not Higher Score:**

See weakness of the paper. Reviewers and AC are concerned about the experiment setting of the paper and the novelty of proposed approach.


**Justification For Why Not Lower Score:**

N/A

**Metareview: Summary, Strengths And Weaknesses:**

The paper proposes an easy-to-implement fine-tuning technique for machine translation via reinforcement learning (RL). Conventionally, the major challenge of using RL in fine-tuning is its large variance in rewards and updates. The author proposed to improve the fine-tuning via 1) conditional mean and variance normalization of the task reward based on a population of conditional samples, sampled from the current (local) policy over a range of temperatures, and 2) importance weight that depends on the mean absolute deviation (MAD) of the probability of conditional samples from the median probability in a sample population. Both techniques aim at reducing variance, improving regularization in the gradients across sets of conditional samples, and mitigating policy collapse. Experiments on multiple machine translation datasets show the effectiveness of the proposed method.



Strength of the paper:
1. This paper is well organized and easy to understand.
2. Experiment results shows proposed method yields promising results and greatly boost stability for RL tuning.
3. Thorough ablations are conducted on multiple datasets and multiple settings, which bring insights for improving RL-Based NMT.

weaknesses of the paper:
1. Experiment settings can be improved in many ways, such as baselines to compare with, implementation of baselines and the conclusion/discussion draw from experiment results. Specifically
    a. Conditional reward normalization (CRN) is identified as the key factor to MAD performance. However, based on the current manuscript (e.g. section 2.2 and discussions elsewhere), it seems that the baseline approaches being compared to such as REINFORCE and MRT, which trivially have conditional variants of their baselines, are NOT conditionally normalized. The importance of CRN is well known, and the comparisons must be "apples to apples".
    b. Techniques such as MRT and MAD rely on sample population statistics for reward normalization, whereas other techniques estimate a conditional baseline and are much more sample-efficient to train, including cited methods in the paper such as in MIXER (learned baseline), self-critical training (MAP sequence estimate defines baseline), and actor-critic methods for sequence prediction (word slot-dependent rewards). Since conditional reward normalization seems to be the dominating factor in performance, it seems essential to also situate the results relative to one or more of these techniques (the former 2 are very easy to try out).
    c. Importance weighting based on how far from the median probability each sequence is in probability seems excessively conservative for noisy models such as text sequence generators, and will certainly foster the stated goal of "exploration-heavy" training, but it will also prevent the model from converging. Since this a heurstic, I feel that some other competing heursitics should be compared to and reported on... For example, perhaps the distance from expected probability is better, it is less "exploration heavy", but will allow the policy to converge (for better or for worse).
    d. It seems that MAD reweighting should be compared with standard regularization baselines, like an entropy term!
2. The main components of the approach 1) mean and variance normalization of conditional rewards, and 2) importance weights based on the distance from the median probability are standard approaches/tricks and are not novel. The improvement of MAD itself seems to be marginal, which makes the title misleading.